# Unity of the Existence of God and the Knowledge of God in the Phenomenological Ontology of Henry

## Weifeng Cui

School of Marxism, Central China Normal University, Wuhan 430079, China; ccnucwf@ccnu.edu.cn

**Abstract:** This study approaches the question of the unity between the existence of God and the knowledge of God. Henry's phenomenology of life, as a phenomenological ontology, offers a phenomenological way to rethink the existence of God and our cognition of God by seeking the essence in life's self-donation. As a phenomenological heritage of Husserl's intentional phenomenology, Henry's phenomenology of self-affection (auto-affection) clarifies the essence of God in the dimension of subjective body or in the flesh. This truth, which presents our absolute immanence, is, in its depth, a divine revelation between God and human. When we experience our own existence in the tonality of corporal life, we receive the existence of God and the knowledge of God.

**Keywords:** life; subjective body; original revelation; immanence; self-affection

## 1. The Absolute Immanence of Life as the Essence of Deity

From the perspective of phenomenology, is "God" transcendent? How does "God" enter into phenomenological discussion? After Eckhart, Michel Henry has realized the radical importance of the original knowledge that is connected with the unity of the immanent life of humanity and God. By clarifying the absolute essence of life, Henry proposes a radical way to rethink our own existence and the existence of God.

In *The essence of manifestation*, as the title indicates, Henry tries to discover the essence of manifestation. The essence here signifies the foundation or the possible condition of the manifestation. The manifestation means the truth, the being unveiled. Henry tries to establish a universal ontological phenomenology, which is an original autonomous ontology, to clarify the signification and the new task of philosophy, and to go beyond the regional philosophy that determines, in an inappropriate way, *the sense of the Being in general*. Henry considers the Being of universal ontological phenomenology as a universal Being, which designates the essence and the object or the content of all the regional ontology. In *The essence of manifestation*, Henry tries to establish an ontology by a real immanent phenomenology that is the phenomenology of life, and this phenomenology is different from Husserl's intentional phenomenology or the historical phenomenology. It is a future phenomenology, which studies the essence of the Being in general. It creates a new phenomenality, which is a "perfect revelation of self-to-self" («révélation parfaite de soi-à-soi» (Laoureux 2005, p. 29). This radical revelation is an experience of self-to-self; it is life's self-affection (auto-affection)[1]. This essence is not a particular individual being (l'étant), it is the universal or absolute Being that designates the individual being and nothingness too. According to him, nothingness is not a *relative nothingness*, which designates the "not Being" of the individual being. It is a real nothingness, "which constitutes the very essence of Being" (Henry 1973, p. 12).

In the first page of the Introduction of *The Essence of Manifestation*, Henry has indicated that his philosophical ambition is to propose a universal ontology that is able to clarify the signification of the Being in general, namely the universal Being. He begins his research by interrogating the signification of the Being of the ego: "Thus should not the true object of an inaugural inquiry be the Being of the ego rather than the ego itself, or more precisely,

the Being in and by which the ego can rise to existence and acquire its own Being? This is why the Cartesian beginning is not at all 'radical', because such a beginning is possible only upon a foundation which he did not clarify and which is more radical than the beginning" (Henry 1973, p. 2).

Henry criticizes the *ego cogito* of Descartes; because it is not an ultimate and absolute foundation, it is still conditioned, and it presupposes an intuition to confirm itself: "Intuition is the foundation of every rational assertion"[2] (Henry 1973, p. 3). The significant intention presupposes the existence of its intended object, namely a transcendental correlation exists between the agent and target of intuition (l'intuitionner et l'intuitionné), in the "intuitionism" of Descartes. According to Henry, "The theory of intuition has already rejected any pretense of reaching all spheres of Being with the type of evidence furnished by the cogito"[3] (Henry 1973, p. 8).

In this sense, the *ego* becomes a real container of the self, which is a regional being, but not a universal Being. There is a difficulty inherent in the singular reality of the *ego*: because the evidence that comes from the singular *ego* is always particular evidence, the sense of the Being in general nevertheless cannot be obtained in the singularity of the *ego*. This is a paradox between individuality and the universality. Cartesian philosophy establishes the relation between the evidence of the knowledge or the truth and the being of the *ego*, but without interrogating the paradox: "the *cogito-consciousness* is not individual, but it is true" (Henry 1973, p. 6). The ultimate task of philosophy is to investigate the universal Being, which is not limited in the *ego cogito*. This task constitutes in "the systematic clarification of being in all its structures and in all its ultimate eidetic differentiations" (Henry 1973, p. 7). The investigation of the privilege of the *ego cogito* presents a "rational phenomenology of the ego", but Henry thinks that this investigation is just particular research, because "the problematic concerning the Being of the ego occupies no more than a strictly limited place in the totality of phenomenological investigations" (Henry 1973, p. 7). Therefore, Henry expands Descartes' investigation of the Being of the ego to an investigation of the universal Being. The philosophical principle of reason is not limited in the *ego cogito*. According to Henry, it should go beyond regional ontology in order to establish a fundamental ontology, which is the *universal phenomenological ontology* but not a formal ontology. It is evident that Henry tries to inverse the traditional relation between formal ontology and material ontology.

According to Henry, there is an original co-belonging between the truth of the ego, which is the self-affective existence of life, and the divine knowledge, which is rooted in the existence of God. In *Philosophy and phenomenology of the body*, Henry proposes an important question: "If self-knowledge is absolute knowledge, the problem surely arises of knowing if the idea of knowledge which God would have of the ego, alongside the immediate revelation of this ego to itself, still has any meaning" (Henry 1975, p. 180). Henry furnishes a response to this question: "in Meister Eckhart, we find an explicit theory of the identity of self-knowledge and divine knowledge" (Henry 1975, p. 181). In *The Essence of the Manifestation*, Henry describes the theory of Eckhart: "the existential union of man with God is possible only on the foundation of their ontological unity" (Henry 1973, p. 309). According to Henry, the more radical contribution of Eckhart consists in clarifying ego's fundamental power of receiving itself and reuniting with itself. We could understand the essence of the deity by this radical unity.

Henry thinks that "it is in self-knowledge where we must read that it is similar to the knowledge of the real being of the ego which is God's knowledge" (Henry 1975, p. 180). In this phrase, "self-knowledge" is an "absolute knowledge" which signifies an original revelation; it is the self-affection (auto-affection) of life. Henry's later philosophy offers a new phenomenological method to discuss the problem of God for us. For him, it is question of showing that in the experience of the *cogito*, there is a more fundamental donation, by which the existence of the *cogito* is possible. In other words, the experience of self presents itself as the possible condition of the experience of the *cogito*, and as the possible condition of the experience of God, which is the Life in capital form. According to Henry,

this experience of the *cogito* presents itself like absolute knowledge, which takes the form of absolute immanence. This absolute immanence is the essence of life that is invisible and refuses all transcendence. To emphasize the tautological form of this knowledge, we could say that it was the self-knowledge of life; it is a pathetic relation without interval, which resides in the absolute immanence of life. In other words, absolute immanence is the essence of life.

In *I Am the Truth: Toward a Philosophy of Christianity*, in order to clarify the radical signification of the immanence, Henry summarizes the double signification of immanence and transcendence:

> What is specific to life as self-revelation is therefore the fact that it reveals itself. This apparent tautology implies two distinct meanings that we must now separate for the first time. Self-revelation, when it concerns the essence of life, means, on the one hand, that it is life that achieves the revelation, that reveals, but on the other hand, that *what life reveals is itself*. And it is here that the mode of revelation specific to life differs fundamentally from that of the world. The world, too, reveals and makes manifest, but within the "outside", casting a thing outside itself, as we have seen, in such a way that it never shows itself as other, different, external, in its setting of radical exteriority that it the "outside-itself" of the world. Hence it is double exterior: external to the power that makes it manifest—and this is where the contrast between Truth and what it makes true intervenes—and also exterior to itself. . . . If, then, Life reveals itself not only in the sense that it achieves revelation but also because it is itself that it reveals in such a revelation, then Life is possible only because its own mode of revelation ignores the world and its "outside". *Living is not possible in the world*. . . . Life embraces, experiences without distance or difference. Solely on this condition can it experience itself, *be itself what it experiences*—and, consequently, be itself that which experiences and which is experienced. (Henry 2003, pp. 29–30)

Immanence exists as the way of affection and immanence as the affection itself[4]. Therefore, in radical immanence, "life is nothing other than that which reveals itself" (Henry 2003, p. 29). Besides, the self-revelation of life is an act of self-revealing, and the Revelation of God is a self-revelation that has nothing to do with transcendence. Absolute immanence or radical immanence at the same time is the fact of affection and the act of affection, which exist as content and form. How can one clarify this absolute immanence and understand the unity of content and form?

Henry thinks that material phenomenology could designate an invisible phenomenological substance: "in the eyes of the material phenomenology, the phenomenological substance is the pathetic immediation in which the life makes the experience of itself" (Henry 1990, p. 7). Life is the principle of all things, "it is a phenomenological life in the sense that the life defines the essence of the pure phenomenality and as a result of the Being" (Henry 1990, p. 7). Affective life that affects itself and feels itself in radical immanence, in the invisible and inexpressible immanence; it is the field of Being and ontology from which the epistemological cognition comes. Henry thinks, "The life hides itself in principle from all the power of the conceivable visibilisation" (Henry 1990, p. 8). Life is the essence of manifestation, and it is radical immanence.

## 2. The Phenomenological Signification of Henry's "Material" in the History of Phenomenology

The concept of "material" in Henry directs closely to the concept of "*hyle*" in Husserl, which marks the radical difference between the phenomenology of Henry and that of Husserl. In *Logical Investigations*, Husserl distinguishes perception (*Wahrnehmung*) from sensation (*Empfindung*). According to him, perception is a complete act of consciousness. We can acquire fundamental intentional donation in perception, and it is a compete act that relates the perceiving subject with the world perceived. The objective donation of the world is accomplished in perception. On the contrary, sensation is not a complete act,

and it is the constitutive element of perception, which means it constitutes the content or the "*hyle*" for perceptive apprehension. It is the pre-donation layer for the perception. If we give a genetic order to the act of consciousness, we can see that sensation is an inferior act to perception. The intuitive donation of the sensation or the being sensed is the beginning of the evidence of our consciousness. Even if Husserl does not consider sensation as a complete act, he should explain how sensation becomes the content of perception and explain how to obtain these sensations. This is the "myth" of the relation between the subject and the world. In fact, it seems that the study of perception in Husserl presupposes a phenomenology of sensation. This is why Henry tries to show that the essence of sensation is the limit of Husserl's *hyletic phenomenology* (Cui 2019). Therefore, Henry wants to establish a material phenomenology to clarify the transcendental essence of the sensation. In the Preface of the *Material Phenomenology* (*Phénoménologie matérielle*), Henry indicates the task of this phenomenology: "radicalizing the question of the phenomenology, it does not simply aim at pure phenomenality; it also interrogates how the phenomenality makes itself a phenomenon. The substance, the stuff, the phenomenological material of which the phenomenality makes—its pure phenomenological materiality. This is the task of the material phenomenality" (Henry 1990, p. 18).

In Henry's eyes, the material, the sensible data or the "hyletic moments of the experience" are non-intentional. They include the *content of the sensation*, for example, the sensual color, the pure impression of sound, and *sentiments* such as pain, joy etc.; they are the experiences defined by their impressionable character. Henry thinks that Husserl does not explain clearly the possibility of the unity of the "*hyle*" and the "form", for Husserl thinks that non-intentional sentiment or sensation does not have the independent status of existence; they are the constitutive part of the intentional act. In Husserl's eyes, it is in the noetic-noematic constitution of objectivity that something could be intentioned and expressed.

At the same time, in the § 11 of the 5$^e$ *Investigation*, Husserl distinguishes the intentional object from the real immanent content in order to explain the difference between the sensorial content, which is the real content for the intentional act, and the intentional object. The first, as non-intentional material, is the real composition of the intentional experience, and it plays, in total or partial manner, the role of fulfillment of the intention.

These so-called immanent contents are therefore merely intended or intentional, while the truly *immanent contents*, which belong to the real make-up (*reellen Bestande*) of intentional experience, are *not intentional*. They constitute the content of the act of consciousness, and provide necessary *points d'appui* which make an intention possible, but themselves are not intended; they are not the objects presented in the act. I do not see color-sensations but colored things, I do not hear tone-sensations but the singer's song, etc. (Husserl 1970, p. 99).

In addition, in the § 14 of the 5$^e$ *Investigation*, Husserl make a phenomenological distinction between the pure sensed or the experienced sensation, which is the real content of an act, and the object of perception, which is the intentional content:

> "I hear" can mean in psychology "I am having sensations": in ordinary speech it means "I am perceiving"; I hear the adagio of the violin, the twittering of the birds etc. . . . whatever the origin of the experienced contents now present in consciousness, we can think that the same sensational contents should be present with a differing interpretation, i.e., that the same contents should serve to ground perceptions of different objects. Interpretations itself can never be reduced to an influx of new sensations; it is an act-character, a mode of consciousness, of "mindedness" (*Zumuteseins*). We call the experiencing of sensations in this conscious manner the perception of the object in question. (Husserl 1970, p. 103)

In *Logical Investigations*, Husserl specifies the objectifying act by the *quality* of the act, and he distinguishes two types of act: the positional act and the non-positional act. The positional act aims at the object as existing; the non-positional act suspends the existence of its object. The latter is based on the former. The non-objectifying act could also be distinguished into the positional mode and the non-positional mode, but in the non-objectifying act, the non-

positional act does not need to be built on the positional act. For example, joy and sadness, as intentional acts, belong to the positional act; they presuppose the existence of the simple object in order to be completely realized. However, we should note that existence here does not signify existence in the consciousness, like the internal object in the consciousness. At the same time, the representation is not an image or a copy of the appearing object, otherwise it presupposes a consciousness of image to explain the constitution of this image-copy of the objective donating intentionally. Briefly, this way of proposing the problem brings about an infinitive regression. In fact, the consciousness of image is another type of act, which has its own essential character and its own quality. Husserl thinks it is absurd to make a distinction between the intentional object and the external physical object. When we say the representation presupposes the existence of an object, it means what exists is the intention, the intended object exists in an intentional way. Consequently, we could not only intend a physical object, but also God, all fictive beings and even a contradictory concept, for example, the circular square. In the intentional consciousness or in the representation, what exists is just the intentional object, regardless of this object's existence or not in the physical world. The notion of representation, which comes from Brentano, is an ambiguous concept that Husserl tries to clarify in *Logical Investigations*, so he replaces it with the term of *noesis* and *noema* in *Idea I*. This terminological change indicates Husserl's transcendental turn in *Idea I*.

In *Idea I*, Husserl thinks, "the sensuous pleasure, pain and tickle sensations, and so forth, and no doubt also sensuous moments belonging to the sphere of 'drives'. We find such concrete really immanent Data as components in more inclusive concrete mental processes which are intentive as wholes; more particularly, we find those sensuous moments overlaid by a stratum which, as it were, 'animates', which bestows sense (or essentially involves a bestowing of sense)—a stratum by which precisely the concrete intentive mental process arises from the *sensuous, which has in itself nothing pertaining to intentionality*" (Husserl 1983, p. 203, Husserl's italicization). Husserl thinks the helytic sensation or the material donation constitutes the noematic sense of the act of consciousness; it means the multiplicity of sensorial donations, which are realized in sensible intuition, which determines the multiplicity of the noematic sense. These sensible data realize the object or the sense of different acts as such; these data form a correlative noematic status. The "material" designates a material region—corresponding to the formal region—of the phenomenological study. In Husserl, the "material" is a region of being that designates the most fundamental relation of the consciousness with the outside world, but it exists in non-intentional manner, and can be informed by the intentional function of the consciousness.

According to Henry, the *hyle* and the *form* are two essential different things; they cannot unit together as a homogenous unity: if two essences are different, they cannot promote the homogeneity together from where all the reality draw their principal possibility.

In Henry, the elucidation of this possibility therefore becomes a central question; it is the question of the reality of absolute subjectivity. The manifestation of the pure impressionable element (the material, the sensible data, the non-intentional sentiment), which is independent from the formalization of the intentionality, becomes a radical question for Henry. Henry proposes a question: is the impressionality of the impression, its activity already accomplished, in itself and by itself, a function of the manifestation, a phenomenological function?

### 3. The Ontological Unity of Reality and Self-Knowledge in the Subjective Body

In *Philosophy and phenomenology of the body*, in order to clarify the ontological essence of the auto-affection of life, Henry develops a phenomenology of body. In *Incarnation: A Philosophy of Flesh,* Henry cited: "Et le Verbe s'est fait chair" (Laoureux 2005, p. 225). According to him, the Christian body indicates a radical turn of the theory of the body in the history of Western philosophy. Henry thinks the fathers try to clarify the essence of the flesh: "Christ had a real body and a real flesh like our own, and that the possibility of salvation takes place in it and in it alone" (Henry 2015, p. 10). However, how can

one understand the flesh of Christianity; in other words, how can one explain the flesh by phenomenology?

In order to clarify the signification of this theme, we should return to the earlier analyses in *Philosophy and phenomenology of the body*. Henry has distinguished three kinds of body:

1. The original being of the subjective body, i.e., the absolute body revealed in the internal transcendental experience of movement. The life of this original body is the absolute life of subjectivity; in it we live, we move, we sense, it is the alpha and the omega of our experience of the world, it is through it that being comes to the world, it is in the resistance which it experiences that the essence of the real is manifested to us and that everything acquires consistency, form, and value.

2. The organic body is the immediate and moving terminus of the absolute movement of the subjective body, or rather it is the ensemble of the termini over which movement has a hold. Because there is a structure to this organic body, it is divided into various transcendent masses whose diversity is always maintained in the unity of the absolute life of the original body. The existence of such structures interior to the organic body is of great importance relative to the problem of internal sensations, a problem about which we have not, as yet, spoken.

3. The objective body which is the object of an external perception and which can become the theme of scientific research is the only body which philosophical tradition knows, and it is this exclusively objective conception which is at the origin of so many false problems—notably the famous problem of the unity of the soul and body—as of so many theories which strive, even though in vain, to resolve them. (Henry 1975, pp. 129–32)

Henry points out: I am the life of my body, the ego is the substance of its organism, the matter and the principle of its movements, and it is because it would be nothing without this foundation. It is the absolute life of subjectivity; our transcendent body finds in it its unity and the principle of ontological determinations. Therefore, Henry's ego is not a pure form without content, but a real substance of the subject manifested in the substantial material of our organic body. Because the organic body confers its unity to the ego, it cannot separate from the subjective reality of our bodily movements. In this sense, the organic body, as a living body, could escape from the phenomenological reduction. According to Henry, the difference between the organic body and the absolute body is a phenomenological difference, but the organic body "presents itself to us in a sort of absolute knowledge. Because it is the strict non-represented correlate of the intentionality of our absolute body, it is always entirely present to us and we possess it in a knowledge which excludes all limitation and all possibility of error" (Henry 1975, p. 194).

Henry studies the duality of the mode by which our body presents itself: the being of our body could be distinguished into an original subjective being, which is the truth of life, and a transcendent being, which is the truth of the world. This double being of the body poses a radical difficulty: why is the body, as a subjective and transcendent being, the only and the same body, and why we can say that the transcendent body is ours? This duality conducts also to this question: how can we know our body from the exterior and from the interior at the same time? According to Henry, the transcendent body is not a real being behind our subjective body, "*it is the real being of the body itself, its absolute being*; it is *the entire being* of this body, a being which is an absolute transparency and in which no element escapes the revelation of original truth" (Henry 1975, p. 119). Consequently, I do not see my body from the exteriority, I am my body, and the being of my body is an absolute immanence. The "I am my body" is more radical than the "I have a body", because the latter supposes an "I" as the ego which possesses a body. The foundation of the affirmation of "I have a body" is that of "I am my body", and the meaning of the "I have" is manifested in the absolute body[5].

The double manifestation of the body implicates a phenomenological dualism, "the latter dualism which, interior to the transcendental relationship of the being-in-the-world, distinguishes that which, in such a relationship, reveals itself interior to a sphere of absolute immanence and, on the other hand, that which manifests itself in the truth of transcendence, is nothing other, as we know, than ontological dualism. If, as Maine de Biran says, the noumenal point of view is derived with respect to the phenomenological point of view, the two forms of dualism which correspond to these two points of view cannot be without relationship: *Cartesian dualism is precisely a deterioration of ontological dualism*" (Henry 1975, p. 150). Therefore, Cartesian dualism meets a radical problem when it tries to explain "the problem of the action of the soul on the body"[6] (Henry 1975, p. 146). Descartes considers the relation of the soul and the body as a causal relation, and uses the idea of the unity of body and soul to resolve this problem. However, the essence of this unity is not clarified in Cartesianism. Descartes uses the notion of the pre-established harmony of God to explain this problem. However, in Henry's eyes, because Descartes proposes this unity out of absolute subjectivity, and he "confused the organic body with the body represented or conceived by the understanding", he cannot clarify the ontological essence of this unity. According to Henry, we must understand the relation between the body and the soul in terms of *a transcendental relation*, or this relation is unintelligible. The body relates necessarily to the external world and situates in the world, which means it is "being in situation". However, the relation of the body to the world is not a transcendent relation, but a transcendental relation. So, considering the body as a subjective body or as an absolute center allows us to understand how the body exists like a zero point in the world. In other words, it is the absolute body which designates originally our relation to the world, and it is on this the transcendent body rests. According to Henry:

> It is not because our body is also a transcendent body, a body such as philosophy understood it before the discovery of the subjective body, that the being of man is a situated being. On the contrary, our objective transcendent body is situated in a strictly determined sense peculiar to it only because our absolute body is once and for all situated as subjectivity in a transcendental relationship with the world. (Henry 1975, p. 193)

Besides, we must indicate that in *Philosophy and phenomenology of the body*, only the subjective body belongs to the immanent sphere, but in *Incarnation*, by integrating the three kinds of body in the absolute immanence or in the original flesh, Henry refines the conception of the flesh[7]. Henry affirms that "*the reality of Christ's body in the Incarnation as a condition for the identification of man with God*" (Henry 2015, p. 9), the reality of Christ's body presents as a sphere of the absolute immanence, which is the essence of life.

### 4. The Problem of Creation in Henry, Lao Tseu and Yangming

*4.1. The Creation of Life and the Creation of World*

In the phenomenology of life and material phenomenology, it seems that Henry tries to find absolute essence, which is the ultimate foundation of the truth and the being. In his eyes, the aim of the ideal phenomenology is not to study the way of which the things present, but the way of which the donation itself give us; it is the way by which the pure manifestation manifests itself, and the way that the pure revelation reveals itself. Therefore, the original object of phenomenology is the way that pure phenomenality "phenomenalizes" itself. The essence of manifestation is different from the manifestation of the world, the essence that Henry seeks is not an essence in the reflexive order or in the intentional order, because the latter is only a particular and determined thing. It is a universal essence, which designates the particular being and the non-being too. Henry considers the essence of being as an absolute essence. In the metaphysical sense, we can compare it with the "Tao" of Lao Tsu[8]. In Lao Tsu's eyes, the "Tao" is the absolute essence, which creates the world. The relationship between *wu* (non-being) and *you/yu* (being) presents the procedure of

the creation of the world. In the *Tao-Te-Ching*, there is a very famous paragraph about the relation of being and non-being:

> The Tao that can be told is not the eternal Tao.
>
> The name that can be named is not the eternal name.
>
> The nameless is the beginning of heaven and Earth.
>
> The named is the mother of the ten thousand things.
>
> Ever desireless, one can see the mystery.
>
> Ever desiring, one sees the manifestations.
>
> These two spring from the same source but differ in name; this appears as darkness.
>
> Darkness within darkness. The gate to all mystery. (Li 1972, chp. I)

It seems that we could also consider the "Tao" as the universal being, which makes the world present. Because there is no philosophy of subjectivity in Lao Tsu, he does not need to deal with the problem of subject-object dualism. However, there is an interesting paragraph in chapter 25, Lao Tsu indicates:

> I do not know its name, Call it Tao.
>
> For lack of a better word, I call it great.
>
> Being great, it flows. It flows far away.
>
> Having gone far, it returns.
>
> Therefore, "Tao is great; Heaven is great; Earth is great; The king is also great."
>
> These are the four great powers of the universe. And the king is one of them.
>
> Man follows Earth.
>
> Earth follows heaven.
>
> Heaven follows the Tao.
>
> Tao follows what is natural. (Li 1972, chp. 25)[9]

If we try to interpret the concept of "Tao" through Henry's phenomenology, we can interpret it as an auto-revelation which is the original being itself, and which is the essence of all things. At the same time, it is invisible in relation to the transcendent appearance of the particular things, but visible in the sense of phenomenological immanence. It seems that the "Tao" was a divine being like the God, which is the most original truth or the absolute essence, and which is the condition or the ground of the creation of the world[10].

In *The essence of manifestation*, Henry indicates:

> It is the positive essence of Being which uncovers itself in the apparently non-essential characteristic of the essence. That such an essence is positive in an ultimate sense is shown in the fact that it is the condition. Everything that is finds in it its foundation. Universal phenomenological ontology, which deliberately orients itself toward the task of an understanding of the essence, is truly fundamental ontology. (Henry 1973, p. 12)

In the above phrase, Henry indicates a relation between the universal ontology and the regional ontology. The absolute essence is just the object of universal phenomenological ontology's research. In Henry, absolute essence is the auto-affection of life or the auto-manifestation of God; in Lao Tsu, it is the way by which the "Tao" creates the world.

In Henry's phenomenological ontology, the manifestation of being is considered as the condition of the fundamental possibility of the essence. In *The essence of manifestation*, Henry indicates: "What does it mean 'to be'? With the determination of the fundamental possibility of the essence as possibility for its arriving itself in itself, it is *the internal structure of the essence*, the original structure of Being itself, which is described and grasped by the problematic" (Henry 1973, p. 275). Henry thinks the internal structure of arriving itself in itself of the essence is not something dead or finished; it is more likely a process of



self-creation which is presented as the auto-affection of life. Therefore, life constitutes an ontological region whose essence is the radical immanence and exists in an auto-affective form. If a radical principle exists for the manifestation of life, it is the auto-affection or the auto-revelation of life. This is an original manifestation, which is not determined by intentional consciousness.

In the above text, we have pointed out the difference between the perception and the sensation in Husserl. Henry criticizes the fact that Husserl's comprehension is still objectivism. However, we should notice that in the fifth investigation of the *Logical Investigations*, Husserl proposes another type of sensation, which is not an intentional sensation, but a state of consciousness affected by itself. When Husserl discusses the problem of genesis of the consciousness, his later works have a close relation with Henry's thoughts. Henry thinks that the sentiment or the sensation could constitute a new region for phenomenological research, especially when we understand the body as an original subjective body. For example, the sentiment of the sufferance caused by hot water does not disappear, even when we withdraw our hands from the hot water. In the immanent sense, the pain caused by the hot water is the passive dimension of the subjectivity, which constitutes the reality of the subjectivity. In this case, this subjectivity is not formal subjectivity, but an individual subjectivity with its concrete content. Using Henry's latter terminology, it is an individual subjective flesh constantly affecting itself and perceiving the world. Life's auto-affection is an original ontological passivity, which is different from the ecstatic passivity in opposition to activity–passivity. In the original ontological passivity, life finds himself and his reality, this is the original truth of the life; Henry calls it the "Pathos" of life, it is God itself.

In Henry's phenomenology, the invisible of life is just its radical immanence and its auto-affection. In other words, the essence of life is its auto-affection. Life, as the living life, is the condition of intentional consciousness; the latter is only a mode of life. The existence without life or pure physical existence, which can only react passively, does not have the capacity to behave actively. Life is the pure experience of the self, which constitutes an original region of the truth. According to Henry, "this truth, that of the truth in its original and universal structure, is also tragic, for it signifies the irremediable and the definitive, the inauguration of an absolute world from which nothing can be taken away, to which nothing can be added, where without detour and without lie, things are what they are, Being is what it is, in this perfect adequateness which is Being itself" (Henry 1973, p. 291).

Because nothing can be taken away from life, and nothing can be added to it, it is the non-differenced "One", the absolute essence. From this idea, we can consider the "same source" of Lao Tsu as the absolute essence, which is not a particular being, but the being of the particular being, and being of the non-being. The "Tao" or the "same source" has its own mode of creation, which is the same as the self-creation of life. However, life here is not only a human life, but also the life of all the living being. God, as the Life, is also a self-creation[11].

When Lao Tsu writes: "Therefore, 'Tao is great; Heaven is great; Earth is great; The king is also great.' These are the four great powers of the universe, and the king is one of them. Man follows Earth. Earth follows heaven. Heaven follows the Tao. Tao follows what is natural", man and the other three great powers constitutes a radical unity, which is the universe. It seems that this idea corresponds to the structure of Heidegger's deity—the Fourfold (*Geviert*) "earth, sky, mortals and divinities"—which indicates the unity between the human and the world (Mitchell 2015). In this fourfold structure of Lao Tsu and Heidegger, a human finds his place and his ontological existential signification in the world. In Lao Tsu, the four great powers find their original existence in the being's auto-affection. In the state of original existence, there is no ontological difference between them, but each one has its individuality. However, when Heidegger discusses the fourfold structure of the deity, it seems that the "earth, sky, mortals and divinities" still presuppose a more radical deity; this being is not a particular being déjà-là, but a being which is the Verb of "making being". In Christianity, it is the "creation" of the world; in Lao Tsu, it is the "creation" of the "ten thousand things": "The Tao begat one. One begat two. Two begat

three. And three begat the ten thousand things" (Li 1972, chp. 42). We could find a similar idea in Henry. According to Henry, it is the Verb, which permits all Selves to live, and it is the "Born" which causes the world "to be". When Henry says that the life and the Christ share "an original co-belonging", it signifies that the life generates God in itself; it is the God itself, as the first "ipseity", is the accomplishment of the life[12].

*4.2. The Immanent Creation of Moral Principle in Wang Yangming*

In the Confucian tradition, there is an important turn in the period of the Song and Ming dynasties. The iconic figure is Wang Yangming[13]. He thinks the principle of the moral consciousness (Liang Zhi, 良知/心) is given by "heaven", which is innate in humans (Wang 1572). It seems that we could consider this innate moral principle as an *a priori* principle, which is not a transcendent principle beyond human cognition (Cui 2019). Could we compare this moral principle with Henry's phenomenological interpretation of God?

In *I Am the Truth: Toward a Philosophy of Christianity*, Henry proposes the identity of the Life with the Truth (Falque 2001). He distinguishes *the truth of the world* and *the Truth of Christianity*: "*the Truth of Christianity differs in essence from the truth of the world*" (Henry 2003, p. 23). According to Henry's analysis, the Truth of Christianity is the condition of the truth of the world. The former is the truth which could reveal itself, and which is the Revelation of God. It is the auto-phenomenalization of pure phenomenality, which refuses all the phenomenality of the world: "*God is that pure Revelation that reveals nothing other than itself. God reveals himself. The revelation of God is his self-revelation*" (Henry 2003, p. 25, Henry's italicization). If the absolute Life is the Life of God, how do we understand the relation between the absolute Life and the human life?

According to S. Laoureux, there are two kinds of auto-affection: auto-affection in the strong sense, which is suitable for the Life of God, and auto-affection in the weak sense, which is suitable for the essence of humanity. In the strong sense, the life affects itself in two ways. On the one hand, life defines its own content through its own affection. For example, the "content" of the joy is this joy itself; on the other hand, the life produces by itself the content of its affection, the content that it is as a self. According to this definition, the life is affected by one content which is itself, and the affecting is the affected. This kind of auto-affection is that of the absolute phenomenological life or God. In contrast, in the weak sense, the I or the transcendental living being draws its own essence from auto-affection. As an I, I affect myself, I am the affecting and the affected. The "self" is the "subject" of this auto-affection and its content. I experience myself, the fact of experiencing myself constitutes my I. However, I do not bring myself into this condition of experiencing myself. I am myself but I am not in this "being-myself", I am experiencing myself without being the source of this experience. The auto-affection, which defines my essence, is not my fact. Moreover, I do not affect myself in an absolute way; in other words, I find myself be affected.

We can propose a question here: why do the later thoughts of Henry imply this distinction? One reason perhaps is that Henry wants to clarify the problem of inter-subjectivity in order to overcome the individualism or solipsism. Henry thinks the living being does not establish himself by himself, there is a Ground (Fond) which is the Life, but this Ground is not different from him; he is the auto-affection in which he affects himself, and himself is just this auto-affection[14].

In Wang Yangming, the moral principle is innate in humans, although sometimes it is concealed by human intentions (Yi, 意), for the intention could be good or bad. Therefore, it must "go beyond" the intention to find the original moral consciousness, which is in essence good (Tu 1976). In this sense, Wang Yangming thinks that we need a practical philosophy to clarify moral consciousness in order to become a complete man or a sacred man (Chang 1955). Consequently, moral consciousness is not only the possible condition of being human, but also an ethical aim to achieve. It is possible for everyone to become a sacred man by corporal and spiritual practice. Furthermore, we can propose a further

question: if there is a mediation—a Son of God—between the absolute Life and the life of human, is there a moral distance between the Son of God and humans?

In the early work of *Le bonheur de Spinoza*, we can find Henry's philosophical disposition was strongly influenced by Spinoza. In the preface of this book, Henry explains his approach in relation to Spinoza, but he also points out his difference with Spinoza. Henry points out that there is a speculative foundation of the religion in Spinoza, but in *I Am the Truth: Toward a Philosophy of Christianity*, he wants to explain the relation of humanity to God in a phenomenological way (Henry 2004a, p. 8; 2004b). According to Henry, we need a material phenomenology to overcome the traditional ontological monism, and this material phenomenology is a phenomenological ontology or a future phenomenology. Henry agrees with Heidegger's idea that Being and God are not identical, but Henry does not agree with the idea that the manifestation of God is produced in the dimension of Being. In contrast, Henry thinks that the manifestation of God, as the original Life, is the basis of the manifestation of Being. In other words, the manifestation of the Life is the possible condition of the manifestation of Being. In addition, the Life does not manifest himself in a transcendent way, but in an immanent way. Therefore, Henry tries to discusses the problem of onto-theology—"how does God enter into philosophy"—by proposing the material phenomenology.

So, it is interesting to compare Henry's interpretation of the manifestation of God with Wang Yangming's interpretation of the "unity of knowing and acting". The two philosophers take a similar philosophical position to develop a new philosophy. In addition, it seems that we also could find a similar way between Henry and Wang Yangming to study the radical relationship between subjectivity and truth, which means returning to the original life to find the original truth, which is not only a theoretical truth, but also an ethical truth. This original truth of life presents itself in an auto-donative or a self-giving way in its ethical existence[15].

## 5. Conclusions

In the history of phenomenology, even in the history of western philosophy, Henry's thoughts are very special. He offers a new philosophical language to expand our philosophical understanding. His "future" material phenomenology not only develops and deepens the phenomenology, but also offers a new immanent phenomenological way to rethink the problem of God. After the publication of *I Am the Truth: Toward a Philosophy of Christianity*, Henry's phenomenology has extended to a phenomenological reading of Christianity. Is the phenomenological reading of Christianity reasonable here? To be sure, Henry has crossed the boundary of classical phenomenology, as Dominique Janicaud has noted that there is a "theological turn" in French phenomenology (Janicaud 1991). Janicaud's critic is reasonable in a sense, but if we begin looking back to past philosophers, we will find that all important philosophers have tried to develop a new method or a new philosophical system to widen or renew our thoughts. To be sure, Henry's phenomenological approach does not presuppose religious faith. It is rather that the development of the phenomenology has led him to rethink the problem of God in a philosophical way, but not in a religious way (Hao 2022).

In the last chapter, we have made a comparison between Henry's phenomenology and two important Chinese philosophers, and this comparison will no doubt bring some difficult problems; for example, whether it will cause some misunderstanding of the three philosophers. After all, they belong to very different traditions and use different philosophical language. However, in this audacious comparison, I would like to demonstrate a new way to understand the different traditions.

**Funding:** This research was funded by [The National Social Science Fund of China] grant number [18CZX050].

**Informed Consent Statement:** Not applicable.

**Conflicts of Interest:** The author declares no conflict of interest.

## Notes

1    In his later works, Henry proposes self-affection (auto-affection) of the flesh (chair), which is a radical immanance of the flesh. It displays the experience of the self as the flesh, which is the experience of the Life of God (Henry 2012).

2    Henry thinks that "The cogito is only one of these rational truths, but precisely because it permits consciousness to attain—at the very heart of its own individual Being—the order of rationality, it remains the ideal of an inquiry which is first realized in itself and to which it imparts or entrusts a specific task: the acquisition of contents which can serve as valid under the rubric of 'truths'" (Henry 1973, p. 12). At the same time, Henry thinks that "The task of philosophy is in no way an accumulation of truths" (Henry 1973, p. 13).

3    In the article «Le commencement cartésien et l'idée de la phénoménologie», Henry analyzes the relation between the philosophy of the *ego cogito* of Descartes and phenomenology: "A une condition et à une seule, à la condition que le *cogito* cartésien constitue l'acte de naissance de la phénoménologie elle-même. En ce cas, une étude phénoménologique du *cogito* n'est pas seulement possible, elle est la seule possible. Que la phénoménologie définisse la seule voie d'accès à ce qui est pensé et doit être pensé dans le *cogito*" (Henry 1997, p. 200).

4    In order to clarify the relationship of the duality of revelation, the self-revelation of life and the revelation of the world, Henry endows an transcendental existence to life. He thinks the self-revelation of life is the possible condition of the revelation of the world, life, as the radical immanance, is the essence of the transcendence or the appearing of world.

5    "I am my body" means: "The original being of my body is an internal transcendental experience and; consequently, the life of this body is a mode of the absolute life of the ego. 'I have a body', this means: A transcendent body manifests itself for me also and presents itself to me as subject to, by a relationship of dependence, the absolute body which, as the theory of the constititution of our own body has shown, also gives basis to this objective body as well as to the relationship of possession which binds it to the ego" (Henry 1975, p. 196).

6    Henry also thinks, "The dualism of the soul and the body, i.e., the original being of the subjective body and the transcendent body, is only a particular case of ontological dualism. The act whereby subjective movement stretches out the hand as an organic mass which it knows interiorly, as the terminus toward which, not its intellectual knowledge, but its motor knowledge transcends itself is no more or less mysterious than the act whereby my look aims at and attains the tree standing there on the hill. The dualism which the description of these phenomena brings to light is not an ontic dualism; it is a dualism which does not differ from the one we recognized between original truth and the truth of transcendent being, and which expresses the relationship fundative of the unity between these two truths, fundative of the unity of experience—it is a dualism which has nothing to do with Cartesian dualism" (Henry 1975, p. 135). From this paragraph, we can know that Henry proposes a new ontological phenomenology to deepen the phenomenological discussion.

7    For a profound analysis of this question (Laoureux 2005, p. 147).

8    Lao Tsu or Laozi (l. c. 500 BCE, Chinese: 老子; pinyin: Lǎozǐ; Wade: Lao³Tzu³), his real name is Li Er ( 李耳, Lǐěr), he was a Chinese philosopher credited with founding the philosophical system of Taoism. He is best known as the author of the Laozi (later retitled the Tao-Te-Ching translated as "The Way of Virtue" or "The Classic of the Way and Virtue") the work which exemplifies his thought. Lao-Tzu—World History Encyclopedia.

9    In this translation, the *Ren* (人) is translated as the *king*, but in the original chinese text, *Ren* means a human being, not the king of a kingdom. From this phrase, we can see that Lao Tseu notices the existence of human in the world, but the human is not a subject in the sense of modern western philosophy, which corresponds to the object, for example the *cogito* of Descartes. The man of Lao Tsu possesses the same ontological status with the *Earth*, *heaven* and the *Tao*.

10    However, in the original sense, the "Tao" is not a religious concept, but a philosophical concept.

11    The comparison between western philosophy and oriental philosophy is a difficult work, because it is easy to misunderstand these two traditions when we try to interpret one by the other. However, it is necessary to make a comparison for discovering new possibilities in thought.

12    In «La répétition de la "philosophie du christianisme"», Tadayoshi Furuso indicates: «c'est pour cette raison—strictement philosophique—que Henry se trouve conduit à poser la Vie absolue qui rend possible l'auto-affection de la vie humaine, laquelle s'avère passive en tant qu'auto-affectée dans l'auto-affection absolue de la Vie, et ne peut s'éprouver soi-même réellement que dans la Vie qui donne la vie à chaque homme. En revanche, la Vie absolue peut elle-même être source de son auto-affection. [ . . . ] Selon Henry, c'est le Verbe de la Vie qui permet à tout Soi vivant, y compris le mien, de venir en soi et de vivre.» (Furuso 2015, p. 88) I agree with Tadayoshi Furuso's idea of the distinction between the absolute life of God, which permits everything to live, and the life of humans. In the main text, when comparing the thought of Henry and the thought of Lao Tsu, we should not forget this radical distinction between God and man in Henry and in Christianity, or we will misunderstand Henry's phenomenology of life as traditional panthesim.

13    Wang Yangming (1472–1529) was a Chinese statesman, general, and Neo–Confucian philosopher. He was one of the leading critics of the orthodox Neo–Confucianism of Zhu Xi (1130–1200). Wang is perhaps best known for his doctrine of the "unity of knowing and acting," which can be interpreted as a denial of the possibility of weakness of will. https://plato.stanford.edu/entries/wang-yangming/, accessed on 9 October 2022.

14     In "Unveiling the Pathos of Life: The Phenomenology of Michel Henry and the Theology of John the Evangelist", John Behr indicates the importance of a new concept of temporality of Henry(the essence of Life's own temporality): there is neither before nor after in the "pathos-filled temporality, but an 'eternal flux in which life continuously experiences itself in the Self that life eternally generates'" (see Behr 2018, pp. 104–26). John Behr's discussion about this new temporality inspires me a lot to consider the difference between God and man: God Himself is the eternal without an end in the time of world, which means the Life of God is the eternal itself, but the life of man has an end in the world. Perhaps this is why there is a distinction between the affection of God and the affection of man: as an I, I affect myself, I am the affecting and the affected. However, I do not bring myself into this condition of experiencing myself. I am myself but I am not in this "being-myself", I experience myself without being the source of this experience. The auto-affection is not my fact. Moreover, I do not affect myself in an absolute way, in other words, I find myself to be affected.

15     By developing a phenomenology of life, Henry presents the phenomenological structure of this ethical existence of man in Christianity and the relationship between man and God. However, Wang Yangming does not have to deal with the problem of creation and the problem of the relationship of man to God. The innate moral principles in Wang Yangming is the principle of value but not the principle of fact. However, in Christianity, the truth of God is not only the truth of value but also the truth of fact. This is the radical difference between Wang Yangming and Henry.

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
