# Peer review of "Unity of the Existence of God and the Knowledge of God in the Phenomenological Ontology of Henry"

_religions, doi:10.3390/rel13100964_

Round 1
Reviewer 1 Report
Well written, this article shows a very good understanding of Michel Henry’s phenomenology of religion and also of the phenomenology of Husserl.
The major problem of this article is its main idea: it does not take into consideration the Christian importance of the difference between God and man, also found in Henry's last thinking. Trying to offer a pantheistic hermeneutic of Henry, the author does not show that this is not present anymore in Henry’s three books on the phenomenology of Christianism. The life does not suspend the Christian idea of the difference between God and man. I think that a pantheistic hermeneutic is not good enough to interpret the Henry's phenomenology of religions.
I recommend to the author some theological reading:
-
Falque, Emmanuel. “Michel Henry Théologien (À Propos De C’est Moi La Vérité).” Laval théologique et philosophique 57, no. 3 (2001): 525. https://doi.org/10.7202/401380ar.
-
Turcan, Nicolae. “Religious Call in Eastern Orthodox Spirituality: A Theo-Phenomenological Approach.” Religions 11, no. 12 (2020): 653. https://doi.org/10.3390/rel11120653 (the first part).
-
Gschwandtner, Christina Maria. Postmodern Apologetics?: Arguments for God in Contemporary Philosophy. Perspectives in Continental Philosophy. New York: Fordham Univ. Press, 2013 (the chapter on Henry).
-
Behr, John. “Unveiling the Pathos of Life: The Phenomenology of Michel Henry and the Theology of John the Evangelist.” Journal of French and Francophone Philosophy 26, no. 2 (2018): 104-26. https://doi.org/10.5195/jffp.2018.861. https://dx.doi.org/10.5195/jffp.2018.861.
Author Response
Thanks a lot for your specific and inspiring suggestion!
For the problem of the static understanding of the duality of appearing, I would like to indicate that Henry points out the relationship of this two modes of revelation, he tries to establish a transcendental relationship between them. He thinks the self-revelation of life is the possible condition of the revelation of the world, and life is the principle of all the things, that means the essence of the transcendence is the immanence. I have added a note (note 15) to clarify this problem in the page 4.
For the problem of the existential dimension of the appearing of Life and its relation to the living. We must notice that Henry never consider the Life as the mode of the living in a traditional way, which makes the dissolution of the individuality in an absolute essence. In contrary, this traditional idea of the pantheistic is just what Henry has criticized in the§35 of the Incarnation: une philosophie de la chair. Henry considers every living as a mode of the Life, but he insists on the individuality of every living things, and the union of the individual with the absolute being does not signify the dissolution of the individual in the absolute. He thinks “life's immanence in every living being thus does not mean that the reality of the human being is dissolved at the same time as its individuality, while (in a phenomenological interpretation that is as decisive as it is novel) the immanent process of absolute Life generates in itself the Ipseity of an original Self as the internal condition of its self-revelation—as the internal condition of its own life and thus of every conceivable life”( see page 181 of the English version of Incarnation). For a better understanding of Henry’s idea, we need to make a distinction between the self-affection in the strong sense and the self-affection in the weak sense (see page 13 of the paper).

Reviewer 2 Report
This article is a very serious and rigorous study of the presuppositions of Michel Henry's radical phenomenology and its relation to an understanding of God more adequate to the dynamisms of human experience and in dialogue with other religious perspectives, which makes it even more valuable.
My assessment from the point of view of the content is the following: although the author shows a clear way to reach or give account on the duality of appearings, his understanding of this duality is nevertheless somehow static. Even the approach to the description of the appearing of Life becomes a little bit schematic, although the nuances of the meanings of self-affection are well pointed out. Nevertheless, I miss some references to the existential dimension of the appearing of Life and its relation to the living, that is, its capacity for alterity (not only in Life but also in the living), the ambiguity under which its revelation takes place, as Michel Henry points out in some passages of "I am the truth" an the paradox of the invisible (Life) surrendering to the visible (world).
From the formal point of view, I have found the following mistakes:
1. Henry with capital letter (page 9).
2. An instead of ab (page 9).
3. Why the laters thoughts of Michel Henry imply this distinction? (page 13).
Author Response
Thanks a lot for your specific and inspiring suggestion, and thanks a lot for offering some helpful articles and books!
I agree with the idea of the difference between God and man, and I have cited S. Laoureux’s distinction between the self-affection in the strong sense and the self-affection in the weak sense to support this idea. Because of lacking a further clarification to this idea in the part of “4.1. The creation of life and the creation of world”, it leads us to understand Henry’s idea as a traditional pantheism. By comparing Henry, Lao Tseu and Wang Yangming, I’d like to give Lao Tseu and Wang Yangming a phenomenological interpretation. Certainly, there is a difference between Lao Tseu and Wang: in a sense, the Dao of Lao Tseu is a transcendental being but not a unique being like God; the principle of the moral consciousness(Liang Zhi, 良知/心) is given by the “heaven”, which is a priori principle innate in the human without beyond human’s cognition.
Meanwhile, I agree with John Boehr’s idea in the article of “Unveiling the Pathos of Life: The Phenomenology of Michel Henry and the Theology of John the Evangelist”: “Henry understood his work as a radicalization or reversal of phenomenology, rather than a turn towards Christanity or the mystical, the supposed ‘theological turn’ in French Phenomenology”(p. 105).
To clarify more clearly my idea in the paper, we need make a distinction between two kinds of auto-affection: the auto-affection in the strong sense, which is suitable to the Life of God, and the auto-affection in the weak sense, which is suitable to the essence of human. The radical difference between God and man lies in God himself is at same time the affecting and the affection, God brings himself in the condition of experiencing himself, God is the absolute ipseity, and himself is just the source of this experience; in contrary, man is not the condition of experiencing himself, man exists in the form of auto-affection, but this existential form is only an act but not a fact (see the page 13 of the paper). So God manifests itself in the form of “le Verbe s'est fait chair” (Jean, l, 14), and God is the original Logos which is the Verb of “make being”. I have added some sentences to the “note 40” for clarifying this idea (see page 12 of the paper). Meanwhile, I have added two notes (note 46 in page 13 and note 48 in page 14) to make my idea more clealy.
